iFit: An intensity-based retrieval for SO<sub>2</sub> and BrO from scattered sunlight 1 2 ultraviolet volcanic plume absorption spectra 3 4 M.R. Burton<sup>1,\*</sup> and G.M. Sawyer<sup>2</sup> 5 6 1. School of Earth and Environment, University of Manchester, Manchester, UK 7 2. Laboratoire Magmas et Volcans, Université Blaise Pascal, Clermont Ferrand, 8 France 9

- 10 \*Corresponding author : mike.burton@manchester.ac.uk
- 11 12

```
13 Abstract
```

14

15 iFit is a new intensity-based retrieval algorithm for direct fitting of measured UV 16 spectra, specifically developed for use in volcanology. It has been designed with a 17 focus on minimising processing of the measured spectra prior to analysis. Here, we 18 report a detailed presentation of the iFit algorithm, and test it in 4 case studies, 19 examining clear sky spectra, SO<sub>2</sub> calibration cell experiments and volcanic SO<sub>2</sub> and 20 BrO retrievals from traverse measurements performed on Mt. Etna volcano, Italy.

21

22 We show that the major source of fit error in the intensity fitting come from air mass 23 factor independent solar spectrum errors, which are, however, easily characterised and 24 removed by dividing the measured spectrum by a pre-calculated solar spectrum 25 residual. Furthermore, we have quantified the magnitude of the flat spectrum in two 26 spectrometers, and shown that this spectrum is strongly spectrometer dependent but 27 temperature independent, opening the possibility for robust analysis and 28 quantification of both SO2 and BrO without the need for temperature stabilisation of 29 the spectrometer.

- We find that iFit can be robustly and easily applied to traverse measurements of 32 volcanic plumes, producing bias-free profiles of SO<sub>2</sub>, and high quality SO<sub>2</sub>/BrO ratios 33 without the need for clear sky background spectra. Fit residuals are typically pure 34 instrumental noise when the residual solar spectrum is removed. 35 We believe that the iFit approach, which avoids the need for a clear sky spectrum and 36 which achieves noise-limited fits, is ideally suited to the automated analysis of spectra 37 produced by networks of scanning UV spectrometers around volcanoes. 38 39 Keywords: SO<sub>2</sub>, volcanic gas, DOAS, UV spectroscopy, iFit 40 1. Introduction 41

The introduction of the USB2000 spectrometer (Galle et al., 2003) revolutionised the 44 measurement of SO<sub>2</sub> fluxes in volcanology. The direct measurement of a spectrum 45 allowed much more sophisticated analysis compared with the "black box" of the 46 previously used COSPEC (Stoiber and Jepsen, 1973). Analysis of USB2000 spectra using differential optical absorption (DOAS) techniques (Platt and Stutz, 2008) 47 48 rapidly became the standard approach for retrieval of SO<sub>2</sub>. The detection and 49 quantification of BrO in USB2000 spectra using the DOAS technique (Bobrowski et 50 al., 2003) further cemented the association between the instrument and the retrieval 51 approach, such that many groups began referring to the instrument by the analysis 52 technique, e.g. mini-DOAS spectrometer. The flexibility of the USB2000 53 spectrometer and associated optics opened the possibility of arrays of scanning 54 spectrometers arranged around volcanoes, permitting automatic, real-time monitoring 55 of SO<sub>2</sub> fluxes. This led to the installation of such networks, first on Montserrat

(Edmonds et al., 2003), then on Etna and Stromboli (e.g., Burton et al., 2009; Salerno
et al., 2009a, b), and later on over 20 volcanoes worldwide as part of the NOVAC
project (Galle et al., 2010).

In this work we present a novel retrieval methodology for volcanic SO<sub>2</sub> and BrO from 61 UV spectra, which we call "iFit" for intensity-fitting. We use an intensity-based 62 approach in which we attempt to directly simulate and fit the measured spectrum as it 63 is, with very limited manipulation of original data. The forward model is capable of calculating a UV spectrum from first principles, following the physical processes 64 65 arising from radiative transfer and instrumental effects which produce the measured 66 spectrum. The main initial motivation for the development of this approach was to 67 avoid the requirement of the DOAS methodology of a measured 'clear-sky' reference 68 spectrum, which can at times be impossible to measure from a fixed array because the 69 plume is so wide that no clear sky spectra are available. Performing a DOAS analysis 70 with a reference spectrum which includes an unknown amount of  $SO_2$  results in an 71 unknown bias in the measured SO<sub>2</sub> amounts; on the contrary, our approach does not 72 require a measured reference spectrum, and therefore we can be confident systematic 73 biases in SO<sub>2</sub> content are not present. In other words, using iFit we can be sure zero is 74 actually zero on our SO<sub>2</sub> scale.

Apart from the advantages of a calculated clear-sky spectrum, there are also several inherent advantages of an intensity-based technique. These include automatic inclusion of the  $I_0$  effect (Aliwell et al., 2002), thanks to a high resolution version of the solar spectrum. The intensity retrieval also allows quantification of the broad absorption produced by the presence of particulates. Furthermore, inclusion of light-

81 dilution effects is direct, and this will be investigated in more detail in a future paper. 82 Finally, as we show in the case studies below, the SO<sub>2</sub> absorption itself has a large 83 broadband component that is removed in the DOAS approach, but which contains 84 useful information that aids in the intensity fitting procedure. 85 86 In the following sections we briefly review the DOAS approach, and describe a 87 hybrid retrieval which is currently in use on Etna and Stromboli scanning networks. 88 We then explain the iFit methodology in detail. We present case studies from clear sky spectra, calibration cell spectra and traverse measurements on Mt. Etna (Sicily, 89 90 Italy). We then discuss the results from the case studies and conclude by examining 91 new frontiers which are opened using the novel retrieval technique. 92 93 2. Volcanic Plume SO<sub>2</sub> and BrO Retrieval Methodologies 94 95 The most widely used volcanic plume SO<sub>2</sub> and BrO retrieval methodologies are based 96 around the DOAS technique (Platt and Stutz, 2008). The DOAS technique consists of 97 taking a measurement spectrum and a clear sky spectrum and dividing them to 98 produce a transmittance spectrum. This is then converted to optical depth by taking 99 the negative natural log of the transmittance. The optical depth spectrum is then 100 normalised by dividing with a smooth polynomial fitted to the optical depth spectrum: 101 102 DOAS spectrum =  $(-\ln(I_m/I_{sky})) / poly[-\ln(I_m/I_{sky})]$ 1. 103 104 Where I<sub>m</sub> is the measured intensity spectrum, I<sub>sky</sub> is the measured clear sky spectrum 105 and *poly*[] is a polynomial fit.

| 106 |                                                                                                         |
|-----|---------------------------------------------------------------------------------------------------------|
| 107 | The resulting spectrum is therefore a differential optical depth spectrum, hence the                    |
| 108 | DOAS name. By applying the same normalisation to calculated optical depth spectra                       |
| 109 | the DOAS spectrum can be simply and, for weak lines, linearly fitted. Differential                      |
| 110 | optical depths significantly greater than 0.1 demonstrate a non-linear relationship with                |
| 111 | increasing gas amount due to Beer's law:                                                                |
| 112 |                                                                                                         |
| 113 | $t = I_m/I_{sky} = exp(-\tau) = exp(-path_amount . absorption coefficient)$ 2.                          |
| 114 |                                                                                                         |
| 115 | where $\tau$ is the optical depth, and t the transmittance. The Taylor expansion of exp(- $\tau)$       |
| 116 | is                                                                                                      |
| 117 | $1 - \tau + (\tau^2/2) - (\tau^3/6) \qquad 3.$                                                          |
| 118 |                                                                                                         |
| 119 | So it is clear that when $\tau$ is less than ~0.1, ( $\tau^2/2$ ) and higher orders are small and exp(- |
| 120 | $\tau)$ approximates to the linear relationship 1- $\tau$ . SO2 abundances in volcanic plumes are       |
| 121 | often significantly greater than 0.1, and therefore more sophisticated techniques are                   |
| 122 | required.                                                                                               |
| 123 |                                                                                                         |
| 124 | In order to accurately calculate optical depth spectra to fit to the measured DOAS                      |
| 125 | spectrum we must first take account of the instrument lineshape (ILS) of our                            |
| 126 | spectrometer, if the resolution of the spectrometer is too low to fully resolve the                     |
| 127 | natural absorption lines, which is typically the case. Since raw spectra are measured in                |
| 128 | intensity, the convolution of the ILS with the measurement must also be performed in                    |
| 129 | terms of intensity. Therefore, the correct procedure for the calculation of optical depth               |

spectra to be fitted to the measured DOAS spectrum is to calculate a transmittance

| 131 | spectrum by multiplying an absorption cross-section of the desired gas with the                  |
|-----|--------------------------------------------------------------------------------------------------|
| 132 | estimated gas amount in the spectrum, convolve this transmittance spectrum with the              |
| 133 | ILS, convert to optical depth by taking the negative natural log and then normalise              |
| 134 | with a fitted polynomial:                                                                        |
| 135 |                                                                                                  |
| 136 | $gas_{\tau} = -\ln(ILS\otimes(exp(-amt.xsec))) / poly[-\ln(ILS\otimes(exp(-amt.xsec)))] 4.$      |
| 137 |                                                                                                  |
| 138 | Where gas_ $\tau$ is the calculated optical depth spectrum to be used in the DOAS fit, $\otimes$ |
| 139 | represents a convolution, amt is the path amount of gas, and xsec is that gas's                  |
| 140 | absorption cross-section.                                                                        |
| 141 |                                                                                                  |
| 142 | It is important to note that:                                                                    |
| 143 |                                                                                                  |
| 144 | $-\ln(ILS\otimes(exp(-amt.xsec.n)) \neq n.(-\ln(ILS\otimes(exp(-amt.xsec)))) $ 5.                |
| 145 |                                                                                                  |
| 146 | Where n is an arbitrary multiplication factor. In other words, when the gas amount               |
| 147 | changes a new transmittance spectrum must be calculated and reconvolved with the                 |
| 148 | ILS; simply multiplying the optical depth spectrum by a factor will not produce the              |
| 149 | same, physically realistic, result.                                                              |
| 150 |                                                                                                  |
| 151 | It is therefore clear that the correct application of the DOAS approach requires that all        |
| 152 | calculations of fitted spectra are performed first in transmittance (intensity) and then         |
| 153 | converted to optical depth each iteration. The question we pose is: if we are                    |
| 154 | performing the calculations of transmittance in terms of intensity, why not perform              |
| 155 | the fit in terms of intensity as well?                                                           |

# 156

Direct fitting of measured intensities is, in fact, frequently used for analysis of UV spectra. The Global Ozone Monitoring Experiment (GOME) is analysed with such a method (Van Roozendael et al., 2012). In their analysis they find significant advantages for the direct fitting approach over DOAS, which neglects the wavelength dependency of the photon path length, producing misfits in ozone retrievals.

We chose the direct fitting approach of iFit for three main reasons. Firstly, it avoids 164 the requirement for a clear sky spectrum, thus greatly enhancing the flexibility and 165 applicability of the retrieval. Secondly, in principle the fewer manipulations (i.e. 166 converting to optical depth) of the raw spectrum the better, as with each manipulation 167 there is the possibility to introduce errors. Finally, since the SO<sub>2</sub> fit is performed on 168 the edge of a strong ozone absorption band there are low intensities at low solar zenith 169 angles. In the DOAS approach this can lead to very large numbers in the DOAS 170 spectrum, as clear sky spectra contain numbers close to zero. In the direct fitting 171 approach there is no such division by small numbers, and the low intensity is 172 manifested as an easy to manage increase in noise levels.

The DOAS retrieval is used both by NOVAC (Galle et al., 2010) and in the recently
published SO<sub>2</sub> retrieval suite by Kantzas et al. (2012). In both cases clear sky spectra
are required, limiting their applicability.

A hybrid direct-fitting DOAS approach was developed for the analysis of UV spectra
collected with the FLAME (Flux Automatic MEasurement) network on Mt. Etna and
Stromboli (Salerno et al. 2009a, b). This worked in differential optical depth, but

- included the Fraunhofer spectrum in order to avoid the need for a clear sky spectrum.
- The retrieval was investigated thoroughly and judged to be suitable for the automatic
- analysis of UV spectra produced by the FLAME networks (Salerno et al, 2009b).

Here we show that a fully direct intensity fit can produce robust results without the need for clear sky spectra. In the following sections we describe the iFit retrieval in detail, and apply it to several case studies, demonstrating its potential in the field of volcanology.

## 190 **3. The Ocean Optics USB2000 and USB2000+ spectrometers**

While the iFit retrieval can in principle be applied to any UV spectrum, it has been 193 developed with a specific focus on analysing spectra produced by the Ocean Optics 194 (www.oceanoptics.com) USB2000 family of spectrometers. These spectrometers are 195 currently the default tool for volcanologists measuring SO<sub>2</sub> fluxes, thanks to their 196 reasonable cost, robustness and ease of use. The spectrometers are illuminated with a 197 fibre optic cable connected to a quartz collimator. Light enters the spectrometer 198 through a vertical slit and illuminates a diffraction grating. The wavelength-sorted 199 light then passes through focussing elements and optional filters before being detected 200 with a 2048-element horizontal CCD. Three elements of this system are of particular 201 relevance for the iFit retrieval. Firstly, the form of the instrument lineshape (ILS) is 202 determined by the entrance slit, and can be modelled through a combination of a 203 Gaussian-boxcar combination, with greater boxcar contribution with increasing slit 204 width. The slit width is fixed when the spectrometer is manufactured, and there are a 205 variety of different widths available, from 25 micron to 200 micron. Accurate

representation of the ILS is essential for producing good quality fits. This is achieved in iFit by directly fitting the width of the ILS during the retrieval, and manually adjusting the relative weight of Gaussian and boxcar forms until optimal fitting is achieved.

The second group of critical instrumental effects that must be addressed are dark 212 current, bias and stray light. From the moment that charges are read out from the CCD, 213 emptying the charge wells, electrons begin to accumulate at a rate which is both pixel-214 dependent and temperature dependent. This signal is called the dark current, and it is a 215 ubiquitous process, continuing whether the CCD is illuminated or not. In typical well-216 illuminated situations the dark current is insignificant (less than 1 part per mil) 217 compared with the measured signals, but in poorly illuminated cases it can become 218 significant. The bias signal is the nominal 'zero' intensity level for the CCD, and is 219 determined by the CCD electronic configuration and temperature. On the USB2000 220 spectrometers the bias level is inversely proportional to temperature, and at high 221 temperatures the bias level may even fall below zero. For a thermo-stabilised 222 instrument the dark current and bias can be removed from measured spectra by 223 subtracting a dark spectrum, which is collected with the same exposure time and 224 number of coadds as measured spectra, but with the CCD unilluminated.

Stray light can be produced by any photons which are scattered within the body of the spectrometer, producing a close-to-flat offset in addition to the bias. This effect can be greatly reduced through the use of a short-pass filter, which eliminates wavelengths greater than those required for SO<sub>2</sub> and BrO retrievals, typically those longer than 400 nm. USB2000's can be ordered with such an optional short-pass filter. In practice, the

effect of stray light is observed as a residual positive offset in parts of the spectrum in which no light should be present, below 300 nm. It is therefore essential to include ~20 nm of the spectrum below 300 nm when specifying the USB2000 spectrometer in order to correctly characterise the stray light signal. It can then be removed by taking the average of a section of the unilluminated spectrum below 300 nm and subtracting this value from the whole spectrum.

The third instrumental effect of note is the flat spectrum, produced by small pixel-to-239 pixel variations in the conversion rate of photons to electrons in the CCD array. We 240 have characterised this effect by illuminating a spectrometer with broadband light 241 from a deuterium lamp (Ocean Optics model DT1000) and dividing the resulting 242 spectrum with a polynomial fit. The amplitude and structure of the flat spectrum 243 varies significantly from one spectrometer to another but for a given spectrometer it 244 appears to be temperature independent (Figure 1). Note that when a measured clear 245 sky spectrum is used in the DOAS approach the flat spectrum is automatically 246 removed. Instead, for iFit it must be accurately characterised, either via direct measurement or inclusion of the retrieval residual. Because it is pixel dependent it is 247 248 not affected by wavelength shifts, and therefore may affect clear sky spectrum 249 corrected DOAS retrievals if the spectrometer temperature changes after the clear sky 250 spectrum is collected.

251

A final ubiquitous characteristic of CCD detectors is noise. There are two main sources of noise, read noise and shot noise. Read noise is produced upon read-out, amplification and analogue-to-digital conversion of the accumulated charge in each pixel of the CCD. Shot noise is an inherent characteristic of the particle nature of light,