# Peer review of "iFit: An intensity-based retrieval for SO2 and BrO from scattered sunlight 1 2 ultraviolet volcanic plume absorption spectra 3 4 M.R. Burton1,\* and G.M. Sawyer2 5 6 1. School of Earth and Environment, University of Manchester, Manchester, UK 7 2. Laboratoire Magmas"

_Atmospheric Measurement Techniques, 2015_

## Referee Comment (RC1) · Anonymous Referee #1 · 24 Feb 2016

**Review of *iFit: An intensity-based retrieval for SO2 and BrO from scattered sunlight ultraviolet volcanic plume absorption spectra* by M. R. Burton and G. M. Sawyer**

February 24, 2016

The manuscript *iFit: An intensity-based retrieval for SO2 and BrO from scattered sunlight ultraviolet volcanic plume absorption spectra* by M. R. Burton and G. M. Sawyer introduces an alternative to the common 'DOAS' fit, which is applied in the spectral evaluation of stray light measurements (here: at volcanoes) in order to quantify trace gas abundances of, in this case, SO2 and BrO.

The new approach uses a high resolution sun spectrum (also called Kurucz spectrum according to [Kurucz et al., 1984]) as its Fraunhofer reference spectrum. As the intensities are calculated at high spectral resolution, this approach does not need explicit corrections of the saturation and $I_0$-effect, which are observed in DOAS evaluations due to the fact that the convolution with the instrument response function does not commute with the exponential function of the Lambert-Beer-Law [Wenig et al., 2005]. Additionally, this approach does not suffer from reference spectra which are potentially contaminated by the volcanic plume above the instrument, which might introduce a variable bias to the observations. This dependence on some 'reference spectra' is however again introduced, as later on specific residual spectra are used to correct the iFit routine.

However, the individual sensitivity of each pixel (via the flat spectrum) needs to be corrected for, as done here in the manuscript. The computational effort to perform the spectral evaluation will be larger due to the convolution operations needed to be performed within the fitting routine.

The topic presented here fits the scope of AMT and as I'm not aware of peer-reviewed publications using a similar approach I would recommend the resubmission of this manuscript after addressing a couple of individual points listed below. The new manuscript could provide information to an even wider audience, as the principle presented is not only restricted to the detection of BrO and SO2, but could be universally applied.

In its current form, the language of this manuscript is fluent and the overall presentation is well structured. It lacks however proper mathematical formulations, carefully designed figures, intercomparison to existing methods and neglects a number of additional points to be considered (instrument line function, Ring effect, ...).

The description of calculations is not sufficiently complete and precise to allow their reproduction by fellow scientists. Furthermore the bibliography of this manuscript is quite limited and does not represent the current knowledge of related measurement principles and field observations.

**1 General comments**

1. It is mentioned, that the standard DOAS approach uses a number of simplifications (which are sometimes corrected for in current literature and could have been cited accordingly), such as wavelength/absorption-dependent air-mass-factors (AMFs) (e.g. [Pukīte et al., 2010], an effect which is not explicitly treated in the iFit approach), assuming commutation of the convolution and exponential function (saturation effect, e.g. [Aliwell et al., 2002, Wenig et al., 2005], corrected iteratively e.g. in [Lübcke et al., 2014]), the $I_0$ effect [Platt et al., 1997] and more. The manuscript contains no direct comparison to the well-established, intercompared and validated DOAS approach ([Roscoe et al., 2010, Aliwell et al., 2002, Kleffmann et al., 2006, Dorn et al., 2013, Xie et al., 2004] and others). A direct comparison would be a substantial (and essential) addition to the current manuscript and could show the advantages of this new approach by comparing the results directly and discussing their differences. The DOAS analysis will be possible, as sufficient SO2 free spectra can be found in the traverse measurements.

2. As this new fitting algorithm is based on a high resolution sun spectrum, it requires a few convolution operations of the high resolution spectra for each fit. This will slow down the evaluation process significantly compared to a typical DOAS fit and could render this algorithm unsuitable for larger datasets, e.g. data from current satellite instruments or multi-year data sets from scanning DOAS networks like FLAME or NOVAC. A more detailed discussion of this aspect giving estimates on the runtime of each of the methods would be appreciated. Even the 'normal' DOAS approach can be too slow, which is why linearisation approaches for future satellite missions (e.g. TROPOMI) have been developed, see e.g. [Beirle et al., 2013].

3. Additionally to the text description of your algorithm, a clean formal mathematical description would be helpful to unambiguously understand each step. This would also help to implement the algorithm.

4. References to possible future publications should be either shortly summarized in order to understand their possible implications or removed. From the current manuscript, it is e.g. not clear how light-dilution effects can be corrected more easily in this approach than in a standard DOAS evaluation.

5. Please try to make precise statements: The current manuscript contains a series of rough estimations and qualitative descriptions: 'We believe', 'low intensities at low SZA', 'very large numbers', 'a few spectra','not surprising' , 'impressively small','relatively broad'. In my personal opinion these words seem to be unsuited for a scientific publication. If you would

precisely describe these quantities, the reader will be able to follow your argumentation more easily and compare your results to his own.

6. An important note about using a residual spectrum to correct for an insufficient spectral retrieval and to explain systematic residual structures: A correct representation of each of the effects considered in the spectral retrieval can be crucial once residual spectra are used to compensate for systematic residual structures, as the residuals linked to different effects add up in the residual used for correction. This spectrum is then scaled by one factor despite the fact that the individual effects scale independently from each other. Therefore the selection of spectra to chose the 'correction residual' from can be important for the stability of the results. This should be discussed more in depth, e.g. a time series of the OD attributed to the remaining residual structure from the fit in relation to SO2 and BrO could be discussed. A method to extract the individually scaling contributions to the residual of the fit from a set of SO2-free spectra can be found e.g. in [Li et al., 2013]. This point is especially important, if certain effects (e.g. the Ring effect, as shown below) are only very roughly approximated.

**2 Specific comments**

1. p2 line 52: Many groups refer to mini-DOAS, as they often used the 'Mini MAX-DOAS' by Hoffmann GmbH, Germany, which was named 'Mini MAX-DOAS'.

2. p3 line 65: 'from radiative transfer': The model presented in this manuscript does use a very basic radiative transfer and does not include any radiative transfer simulations. I would therefore suggest to change this phrase to 'from strongly simplified radiative transfer', or 'DOAS'-like approximation of radiative transfer, as radiative transfer effects such as Rayleigh and Mie scattering are also just effectively treated by a polynomial depending on wavelength. Thus the only difference of the iFit algorithm described in this paragraph to typical current DOAS evaluations is the use of a convoluted solar atlas as Fraunhofer reference and the calculation at high spectral resolution. This has been so far not used to calculate column densities of absorbers, but only to obtain the wavelength calibration of measured spectra (e.g. in WinDOAS, DOASIS, and other 'so called' Kurucz-fits prior to the 'real' fit to obtain the column densities of the respective absorber). Note that this approach is however used e.g. in the IR-CO2 community for some time (see e.g. [O'Dell et al., 2012]).

3. p3 line 67 and throughout the manuscript: The DOAS method typically needs no 'clear sky spectrum' for volcanic observations, but a 'volcanic plume-free' spectrum.

4. p3 l113ff: Because there is a non-linear relation, $\tau$ is typically calculated from $-\ln(I_m/I_{sky})$ and not from $I_m/I_{sky}$. Then the mentioned Taylor extension is not needed at this point. The problem at optical depths of 0.1 is typically another one, and that is, that the linear operation of convolution and the exponential function do not commute (e.g.

[Wenig et al., 2005, Aliwell et al., 2002]): This leads to the so called saturation effect, which is well known and often iteratively corrected for, if necessary ([Lübcke et al., 2014]) for the relevant absorbers. For a broadband absorber, this effect might be often negligible, but especially for line absorbers (e.g. water vapour), this effect can lead to significant differences.

This effect needs to be separated from the change of effective lightpath which is induced by strong absorptions: To quantify these problems, dedicated radiative transfer calculations are needed, as e.g. the so called correlated-k method applied to absorbers such as H2O, CH4 and O2 in the near IR [Buchwitz et al., 2000] for atmospheric stray light measurements. Such a model is not applied here.

5. p4 l102: The polynomial is typically not fitted separately to the actual DOAS fit, but simultaneously with all absorbers. Other filter approaches (e.g. using binomial filters of fourier transform filters) exist.

6. p6 l144ff: As pointed out above, this is often called 'Saturation effect'. Existing literature could have been cited. To circumvent calculation of high-resolution spectra to intrinsically incorporate the saturation effect, the saturation-corrected spectra are often stored in a look-up-table (e.g. [Lübcke et al., 2014]) or the effect can be often linearised within a certain range of observed slant column densities and represented by two effective convoluted cross-section or a Taylor expansion [Pukīte et al., 2010]. A third method was employed in [Wagner et al., 2003], which rescales the obtained column densities obtained from a fit with a non-corrected cross-section at low resolution.

7. p7 l157ff: Here a few aspects seem to be mixed up. The iFit algorithm also does not take into account the wavelength dependence of the photon path length, as the absorption cross-section is also only multiplied by one scalar, instead of correctly simulating radiative transfer processes for each wavelength (which would require a radiative transfer model). Fitting in intensity space does not make a difference here. DOAS does not do this either, only if further corrections are applied.

8. p7 l165: iFit might require a few less manipulations, as fewer issues such as I0 and saturation correction needs to be tackled explicitly. However, the computational effort is significantly larger. This needs to be discussed.

9. p7 l169ff: In the direct fitting approach there might not be divisions by small numbers (whatever that is), but a larger signal, from which another large signal (ozone absorption e.g.) needs to be subtracted. In terms of error propagation this will lead to the same result, as one side can be transposed to the other by a simple logarithm/exponential function.

10. p7 l178ff: As previous publications [1] the publication by [Salerno et al., 2009] contains some significant technical mistakes. It uses different instrument function widths for the convolution of different (pseudo-) absorbers in the fit. As only one instrument is used, there is only one instrument function
* * *
[1]See e.g. http://www.atmos-chem-phys-discuss.net/10/C2766/2010/acpd-10-C2766-2010.pdf

(for a given configuration, temperature, ...). It is therefore questionable how well results can be reproduced using a correct spectral retrieval. It is however true that it uses a convoluted spectral solar atlas as reference spectrum.

11. p8 l202: The shape of the ILF is determined by the combined effects of the fibre used, the slit, the grating and imperfections of the mirrors of the spectrometer.

12. p9 l207: Why was the ILF manually determined and not fitted within iFit or separately determined using a spectrum of a line emitter? (or is it fitted, as suggested in the BrO fit settings on p24 l581 'FWHM'? This remains unclear and needs to be clarified.) How well does this parametrization represent the real instrument function? Is it kept constant for different spectrometer temperatures?

13. p9 l211ff: The readout and correction procedure for CCDs and CMOS detector signals are typically similar for various types of applications, ranging from astronomy over spectroscopy to photography. The fact that instrumental parameters change with temperature are known and published and could have been cited. Spectrometer specific properties can then still be described, if necessary.

14. p10 l231ff: As stray-light is present in virtually all spectrometers, it is typically to some amount also corrected for by a fitted intensity offset (or inverse Fraunhofer reference for a constant offset) in recent publications aiming at the detection of optical depths in the range of several $10^{-4}$. This could have been cited and is ideally suited to be implemented in intensity space!

15. p10 l238: Typical DOAS applications avoid the need for a flat spectrum by dividing two measured spectra from the same instrument. In this case, it is however necessary and good that this was done and properly described. It is however no new effect and is also regularly applied in Astronomy and Photography. For hyperspectral satellite instruments this is done in the radiometric calibration step, see e.g. [Munro et al., 2015] and references therein. For GOME2 data, the pixel-to-pixel gain calibration data is obtained from LED spectra.

16. p10 l252ff: The used types of noise are not all defined, especially not 'instrumental noise', which dominates apparently the residuals. I hope this is a confusion and the residuals are dominated by photon shot noise (same error on p27 l676). From my experience various USB2000 spectrometer are only limited by shot noise at short total exposure times for RMS of the residuals of some $10^{-3}$ and larger, especially if large amounts of instrumental stray light are observed. Unfortunately the residuals cannot be visually examined from figures 4+8+13 due to the choice of y-axis intervals.

17. p12 l281: See comment to p3 line 65.

18. p12 l288: please cite [Kurucz et al., 1984, Chance and Kurucz, 2010] correctly.

19. p12 l297ff: 'identical', 'acceptable speed': Please quantify.

20. p13 l302 The Ring effect does not only depend on the SZA, but also strongly on the relative solar azimuth angle due to its characteristic phase function, see e.g. [Wagner et al., 2009] and references therein.

21. p13 l304ff The Ring-effect due to rotational Raman scattering (RRS) is by now well known and discussed in literature in depth: [Grainger and Ring, 1962] [Brinkmann, 1968] [Bussemer, 1993] [Fish and Jones, 1995] [Chance and Spurr, 1997] [Langford et al., 2007] and [Wagner et al., 2009] to name just a few publications show different aspects of this effect and even contributions of vibrational Raman scattering were detected in measured spectra [Lampel et al., 2015], which won't play a role here, as it leads to a red-shift of the observed sun spectrum. It is also implemented in state-of-the-art-RTM, e.g. [Rozanov and Vountas, 2014]. Therefore this effect can be effectively compensated for. Using an inverse Fraunhofer spectrum in order to compensate for the contribution of RRS in scattered light spectra was done in the 80's (e.g. [Noxon et al., 1979]), but currently these processes are better understood and can thus be better compensated for.

    Using the inverse of a Fraunhofer spectrum will lead to an overestimation of the Ring effect at shorter wavelengths, as the 'redistribution' of incoming photons takes place only within a few nm, thus the absolute intensity of the Raman signal itself at shorter wavelengths is also smaller. This is shown in Figure 1. As the Ring effect is caused by inelastic scattering on N2/O2 molecules, most of the Ring signal originates from the lower troposphere, below the main ozone absorptions. This needs to be considered when calculating the Ring spectrum and is the reason why a measured spectrum at ground level serves often as a good approximation to calculate the Ring spectrum from.

    For the publication by [Chance and Spurr, 1997] the corresponding RRS response for a high resolution solar atlas can be found and is regularly used (e.g. [Roscoe et al., 2010]). Also software packages (e.g. DOASIS, [Kraus, 2006]) include routines to calculate RRS spectra, which are basically convolutions with the spectral Raman response at the corresponding wavelength including the overall wavelength dependence of the inelastic scattering itself (for details see e.g. [Chance and Spurr, 1997] or [Lampel et al., 2015], where also illustrations can be found.).

    iFit operates mostly in intensity space. Scattering due to RRS modifies intensities rather than optical depths and would be therefore suited to be directly implemented in an intensity-based fit instead of using its approximation in a DOAS fit by Taylor expansion. This is done e.g. in [Chance and Spurr, 1997].

    As the Ring spectrum can, also also stated in the manuscript, contribute with several percent to the intensity of the observed spectrum, I would like to suggest to implement this effect more in detail instead of using very rough approximations.

22. p14 l346: Why is it the solar spectrum, which causes these residuals? This would have serious implications for all algorithms using these spectra

[Figure]

Figure 1: Overview over arbitrarily scaled spectra at 0.6nm spectral resolution of: sunlight, the corresponding Raman spectrum, an inverse sun spectrum and the Ring spectrum calculated by DOASIS.

for wavelength calibration (various satellite instruments, QDOAS calibration routine, ...). [Wagner et al., 2015] suggested a method or radiometric calibration and then compared their sun spectrum to a radiometrically calibrated spectrum. Looking at the quantum efficiency of the detector used in your spectrometer (shown in the data sheet), this could be the reason, as the total quantum efficiency of DOAS spectrometers is often not quantified. How did you exclude this?

23. p14 l348: How were these 'few spectra' selected from the observations? (Compare also: [Li et al., 2013]) In the end this selection of residual spectra is somewhat equivalent to choosing a 'clear sky reference', where is the actual advantage in this case?

24. p14 l351: How was 'pure random noise' identified? At which OD it was pure noise? When averaging pure random noise it should behave like $\sqrt{\frac{1}{n}}$, was this observed? Observations using spectra zenith sky spectra with a total exposure time of several seconds on a USB2000 can yield residual spectra of only a few $10^{-3}$ when evaluated against each other, why is the residual here often significantly larger? Does that show that the residual is not only 'pure random noise' ?

25. p14 332ff: The ILS is parametrized by a sum of a boxcar and a Gaussian function. How was this motivated? Are real instrument functions sufficiently well represented by this parametrisation? A figure comparing a measured instrument function (from a measured spectrum of a line

source) and the corresponding parametrized representation could illustrate the differences. It is said, that the weighting and width of the two functions were 'found by inspection': How can this been done by somebody applying your iFit? Why have these few parameters not been added to the overall fitting routine? Why was not a measured instrument function used?

26. p15 l366ff: The fit presented here is most probably an over-determined system of equations, as more individual detector channels than unknowns to be determined exist. The manuscript states that for iFit an 'optimal estimation approach' is used. This is typically applied when the problem to be solved is ill-posed/underdetermined - in this case 'optimal estimation' is often applied; an a-priori is used to find a solution. Why was this applied here, what is the a-priori, which cost-functions need to be minimized by the Levenberg-Marquardt algorithm? How do the resulting averaging kernels look like? Why would a ordinary non-linear least-squares fit using Levenberg-Marquardt not be sufficient?

27. p15 l366ff: The whole paragraph lacks exact formulation of the mathematical formulae used.

28. p16 l382ff: The whole paragraph lacks citations and exact formulation of the mathematical formulae used. It could be completely removed and replaced by one citation of literature about the LM method.

29. p16 l397: A new K must be calculated, but actually is not. Please clarify and add an error estimate for the use of the empirical, simplified approach.

30. p17 l404ff: With Levenberg-Marquardt the error is typically determined from the size of the residual and the inverse of the covariance matrix of the model. Why was a 3-element boxcar smoothing applied before determining the size of the residual? This is some kind of high-pass filter - but why are broad band structures filtered out? This would neglect broad band structures present in the residual. As seen from figure 5 (which is not referenced), this leads to smaller numbers. But what is the intention of applying this filter? This could be motivated more clearly. (Again, missing formulae)

31. p18 l438ff: Without short-pass filter?

32. p19 l473f: Also the ILF changes significantly with temperature, as reported by G. Pinardi et al in 2007 in a NOVAC report. Given the large overlaying ozone absorption in the SO2 evaluation range, this will then still favor a temperature stabilized version of the spectrometer. Has this effect been studied, maybe together with the temperature dependence of the flat spectrum, which is shown in the manuscript?

33. p18 l438: What are 'optimal intensity levels'? Has the inherent non-linearity of the response of the spectrometer been considered (see e.g. http://oceanoptics.com/wp-content/uploads/OOINLCorrect-Linearity-Coeff-Proc.pdf )?

34. p24 l587: Please put these findings in relation with previous studies at Mt. Etna, e.g. [Bobrowski and Giuffrida, 2012] and references therein.

35. p24 l593: The discussion of the temperature dependent ILF is again missing, which at least to some extend renders the conclusion that not temperature stabilization is needed any more, obsolete. From my experience passive MAX-DOAS measurements during daylight are rarely dark-current limited.

36. p24 l599: This is also the case for DOAS approaches: If the broad-band part of the absorbers are not filtered out (as often the case for passive DOAS applications), the remaining residual will also provide these information. (p28 l685ff also)

    From this paragraph this argument is not clear why aerosol and ash extinction and light dilution can be efficiently and reliable implemented.

37. Figure 5: Units are missing or at least the fit error should be ppm·m?

38. Figure 6: Additional measurement errors would show if the variations from measurement to measurement are indeed correctly characterized by the measurement error. Please use the same units in all plots, if possible either ppm·m or molec cm$^{-2}$.

39. Figure 8+13: Please rescale the residual spectra to a reasonable magnitude, so that its size can be estimated directly and not indirectly from the fits of the absorbers.

40. Figure 9: The plots shown here to visualize the variation of the fit result with changing fitting wavelength interval could be seen as one-dimensional subset of the plots produced by the method introduced in [Vogel et al., 2013].

41. Figure 9+10: In figure 9 the SO2 values decrease for decreasing start wavelength for both gas cells. Also the straylight contribution will increase, due to lower sunlight intensity. In figure 10 the values decrease during the afternoon, when the sunlight is turning more red with time. A larger relative intensity of red/infra-red light in scattered sunlight could also increase the instrumental stray-light signal, as several short-pass filter which block light in the blue and green spectral region do not block light in the near IR. Assuming the filter OF1-U325C to be a Hoya U325C filter, it is transmitting in the NIR, thus it might point towards an instrumental stray-light problem of the setup? Could a separation of the instrumental stray-light correction (e.g. constant intensity offset) and the Ring effect (using a better approximation of the real RRS signal), as done for several current trace-gas retrievals, improve the results?

42. Figure 12+14 could be combined in one plot for simplicity and readability.

43. Figure 14: The BrO slant column densities shown here were calculated from spectra recorded in the early morning of January 7th, 2007, shortly after sunrise. Looking at the magnitude of the baseline values between a SZA of $\approx 83$ to $70°$, these could be caused by stratospheric BrO absorption. From the manuscript it is not clear how the contribution of stratospheric BrO absorption has been corrected for. The stratospheric BrO VCD could

be determined from a Langley plots using spectra with low SO2 values or it is found in literature. In a correlation with SO2 dSCD this would not cancel out, as it's not constant over time.

The plot could be extended by adding the individual measurement errors as grey background, in order to visualize the relation between the variation of slant column densities within a short period of time and the measurement error.

**3    Technical comments**

1. Please use commonly used symbols in the equations throughout the manuscript, instead of full words. This simplifies reading the manuscript. E.g. scattering or absorption cross-section values are typically denoted by $\sigma(\lambda)$ in different fields of physics. Also please use commonly used notations for numbers, e.g. $5e^{16} \neq 5e16$, use instead $5 \times 10^{16}$

2. Please annotate each of the axis in your plots with the corresponding quantity and its unit, if possible.

**References**

[Aliwell et al., 2002] Aliwell, S. R., Van Roozendael, M., Johnston, P. V., Richter, A., Wagner, T., Arlander, D. W., Burrows, J. P., Fish, D. J., Jones, R. L., Tørnkvist, K. K., Lambert, J.-C., Pfeilsticker, K., and Pundt, I. (2002). Analysis for bro in zenith-sky spectra: An intercomparison exercise for analysis improvement. *Journal of Geophysical Research: Atmospheres*, 107(D14):ACH 10–1–ACH 10–20.

[Beirle et al., 2013] Beirle, S., Sihler, H., and Wagner, T. (2013). Linearisation of the effects of spectral shift and stretch in DOAS analysis. *Atmospheric Measurement Techniques*, 6(3):661–675.

[Bobrowski and Giuffrida, 2012] Bobrowski, N. and Giuffrida, G. (2012). Bromine monoxide / sulphur dioxide ratios in relation to volcanological observations at mt. etna 2006-2009. *Solid Earth*, 3(2):433–445.

[Brinkmann, 1968] Brinkmann, R. (1968). Rotational raman scattering in planetary atmospheres. *The Astrophysical Journal*, 154:1087.

[Buchwitz et al., 2000] Buchwitz, M., Rozanov, V. V., and Burrows, J. P. (2000). A correlated-k distribution scheme for overlapping gases suitable for retrieval of atmospheric constituents from moderate resolution radiance measurements in the visible/near-infrared spectral region. *Journal of Geophysical Research: Atmospheres (1984–2012)*, 105(D12):15247–15261.

[Bussemer, 1993] Bussemer, M. (1993). Der Ring- Effekt: Ursachen und Einfluß auf die spektroskopische Messung stratosphärischer Spurenstoffe. Diploma thesis, Heidelberg University, Heidelberg, Germany.

[Chance and Kurucz, 2010] Chance, K. and Kurucz, R. (2010). An improved high-resolution solar reference spectrum for earth's atmosphere measurements in the ultraviolet, visible, and near infrared. *Journal of Quantitative Spectroscopy and Radiative Transfer*, 111(9):1289 – 1295. Special Issue Dedicated to Laurence S. Rothman on the Occasion of his 70th Birthday.

[Chance and Spurr, 1997] Chance, K. and Spurr, R. J. D. (1997). Ring effect studies; Rayleigh scattering, including molecular parameters for rotational Raman scattering and the Fraunhofer spectrum. *Appl. Opt.*, 36:5224–5230.

[Dorn et al., 2013] Dorn, H.-P., Apodaca, R. L., Ball, S. M., Brauers, T., Brown, S. S., Crowley, J. N., Dubé, W. P., Fuchs, H., Häseler, R., Heitmann, U., Jones, R. L., Kiendler-Scharr, A., Labazan, I., Langridge, J. M., Meinen, J., Mentel, T. F., Platt, U., Pöhler, D., Rohrer, F., Ruth, A. A., Schlosser, E., Schuster, G., Shillings, A. J. L., Simpson, W. R., Thieser, J., Tillmann, R., Varma, R., Venables, D. S., and Wahner, A. (2013). Intercomparison of no3 radical detection instruments in the atmosphere simulation chamber saphir. *Atmospheric Measurement Techniques*, 6(5):1111–1140.

[Fish and Jones, 1995] Fish, D. and Jones, R. (1995). Rotational Raman scattering and the ring effect in zenith-sky spectra. *Geophysical Research Letters*, 22(7):811–814.

[Grainger and Ring, 1962] Grainger, J. and Ring, J. (1962). Anomalous fraunhofer line profiles. *Nature*, 193:762.

[Kleffmann et al., 2006] Kleffmann, J., Lörzer, J., Wiesen, P., Kern, C., Trick, S., Volkamer, R., Rodenas, M., and Wirtz, K. (2006). Intercomparison of the {DOAS} and {LOPAP} techniques for the detection of nitrous acid (hono). *Atmospheric Environment*, 40(20):3640 – 3652.

[Kraus, 2006] Kraus, S. (2006). *DOASIS - A Framework Design for DOAS*. Dissertation, Heidelberg University.

[Kurucz et al., 1984] Kurucz, R. L., Furenlid, I., Brault, J., and Testerman, L. (1984). *Solar Flux Atlas from 296 to 1300 nm*. National Solar Observatry, Sunspot, New Mexico, U.S.A.

[Lampel et al., 2015] Lampel, J., Frieß, U., and Platt, U. (2015). The impact of vibrational raman scattering of air on doas measurements of atmospheric trace gases. *Atmospheric Measurement Techniques*, 8(9):3767–3787.

[Langford et al., 2007] Langford, A. O., Schofield, R., Daniel, J. S., Portmann, R. W., Melamed, M. L., Miller, H. L., Dutton, E. G., and Solomon, S. (2007). On the variability of the Ring effect in the near ultraviolet: understanding the role of aerosols and multiple scattering. *Atmos. Chem. Phys.*, 7(3):575–586.

[Li et al., 2013] Li, C., Joiner, J., Krotkov, N. A., and Bhartia, P. K. (2013). A fast and sensitive new satellite SO2 retrieval algorithm based on principal component analysis: Application to the ozone monitoring instrument. *Geophysical Research Letters*, 40(23):6314–6318.

[Lübcke et al., 2014] Lübcke, P., Bobrowski, N., Arellano, S., Galle, B., Garzón, G., Vogel, L., and Platt, U. (2014). BrO/SO$_2$ molar ratios from scanning doas measurements in the novac network. *Solid Earth*, 5(1):409–424.

[Munro et al., 2015] Munro, R., Lang, R., Klaes, D., Poli, G., Retscher, C., Lindstrot, R., Huckle, R., Lacan, A., Grzegorski, M., Holdak, A., Kokhanovsky, A., Livschitz, J., and Eisinger, M. (2015). The gome-2 instrument on the metop series of satellites: instrument design, calibration, and level 1 data processing – an overview. *Atmospheric Measurement Techniques Discussions*, 8:8645–8700.

[Noxon et al., 1979] Noxon, J. F., Whipple, E. C., and Hyde, R. S. (1979). Stratospheric no2: 1. observational method and behavior at mid-latitude. *Journal of Geophysical Research: Oceans*, 84(C8):5047–5065.

[O'Dell et al., 2012] O'Dell, C., Connor, B., Bösch, H., O'Brien, D., Frankenberg, C., Castano, R., Christi, M., Eldering, A., Fisher, B., Gunson, M., et al. (2012). The acos co2 retrieval algorithm-part 1: Description and validation against synthetic observations.

[Platt et al., 1997] Platt, U., Marquard, L., Wagner, T., and Perner, D. (1997). Corrections for zenith scattered light DOAS. *Geophys. Res. Letters*, 24(14):1759–1762.

[Pukīte et al., 2010] Pukīte, Janis, J., Kühl, S., Deutschmann, T., Platt, U., and Wagner, T. (2010). Extending differential optical absorption spectroscopy for limb measurements in the UV. *Atmospheric Measurement Techniques*, 3(3):631–653.

[Roscoe et al., 2010] Roscoe, H. K., Van Roozendael, M., Fayt, C., du Piesanie, A., Abuhassan, N., Adams, C., Akrami, M., Cede, A., Chong, J., Clémer, K., Friess, U., Gil Ojeda, M., Goutail, F., Graves, R., Griesfeller, A., Großmann, K., Hemerijckx, G., Hendrick, F., Herman, J., Hermans, C., Irie, H., Johnston, P. V., Kanaya, Y., Kreher, K., Leigh, R., Merlaud, A., Mount, G. H., Navarro, M., Oetjen, H., Pazmino, A., Perez-Camacho, M., Peters, E., Pinardi, G., Puentedura, O., Richter, A., Schönhardt, A., Shaiganfar, R., Spinei, E., Strong, K., Takashima, H., Vlemmix, T., Vrekoussis, M., Wagner, T., Wittrock, F., Yela, M., Yilmaz, S., Boersma, F., Hains, J., Kroon, M., Piters, A., and Kim, Y. J. (2010). Intercomparison of slant column measurements of $NO_2$ and $O_4$ by max-doas and zenith-sky uv and visible spectrometers. *Atmospheric Measurement Techniques*, 3(6):1629–1646.

[Rozanov and Vountas, 2014] Rozanov, V. V. and Vountas, M. (2014). Radiative transfer equation accounting for rotational raman scattering and its solution by the discrete-ordinates method. *Journal of Quantitative Spectroscopy and Radiative Transfer*, 133(0):603 – 618.

[Salerno et al., 2009] Salerno, G., Burton, M., Oppenheimer, C., Caltabiano, T., Tsanev, V., and Bruno, N. (2009). Novel retrieval of volcanic so 2 abundance from ultraviolet spectra. *Journal of Volcanology and Geothermal Research*, 181(1):141–153.

[Vogel et al., 2013] Vogel, L., Sihler, H., Lampel, J., Wagner, T., and Platt, U. (2013). Retrieval interval mapping: a tool to visualize the impact of the spectral retrieval range on differential optical absorption spectroscopy evaluations. *Atmospheric Measurement Techniques*, 6(2):275–299.

[Wagner et al., 2015] Wagner, T., Beirle, S., Dörner, S., Penning de Vries, M., Remmers, J., Rozanov, A., and Shaiganfar, R. (2015). A new method for the absolute radiance calibration for uv–vis measurements of scattered sunlight. *Atmospheric Measurement Techniques*, 8(10):4265–4280.

[Wagner et al., 2009] Wagner, T., Deutschmann, T., and Platt, U. (2009). Determination of aerosol properties from MAX-DOAS observations of the Ring effect. *Atmospheric Measurement Techniques*, 2(2):495–512.

[Wagner et al., 2003] Wagner, T., Heland, J., Zöger, M., and Platt, U. (2003). A fast $h_2o$ total column density product from gome–validation with in-situ aircraft measurements. *Atmospheric Chemistry and Physics*, 3(3):651–663.

[Wenig et al., 2005] Wenig, M., Jähne, B., and Platt, U. (2005). Operator representation as a new differential optical absorption spectroscopy formalism. *Applied optics*, 44(16):3246–3253.

[Xie et al., 2004] Xie, P., Liu, W., Fu, Q., Wang, R., Liu, J., and Wei, Q. (2004). Intercomparison of no x, so2, o3, and aromatic hydrocarbons measured by a commercial doas system and traditional point monitoring techniques. *Advances in Atmospheric Sciences*, 21(2):211–219.

---

## Referee Comment (RC2) · Anonymous Referee #2 · 16 Mar 2016

Review of

**iFit: An intensity-based retrieval for SO$_2$ and BrO from scattered sunlight ultraviolet volcanic plume absorption spectra**

**by M.R. Burton and G. M. Sawyer**

**General Remarks**

This manuscript describes a new method for retrieving column densities of certain atmospheric trace gases from the spectral intensity of scattered ultraviolet radiation falling through them. It is very similar to the well-established technique of Differential Optical Absorption Spectroscopy (DOAS), but differs slightly in that the spectral measurements are described by a forward model in intensity space rather than optical depth space. As is the case for DOAS fits, the model parameters are iteratively adjusted in a fit routine until the best match between forward model and measurement is obtained. The parameters for which this best fit is obtained are considered the best estimate of the atmospheric state, which includes the trace gas column densities.

There are several novel and very innovative ideas presented in this paper. For one, the authors show that, for spectrometers typically used in volcanic gas measurements, the sensitivity of the individual CCD pixels to incident radiation varies by several percent. This is not unexpected but has not been studied in great detail because it was thought that this effect cancels out in conventional DOAS fits because the spectra are always compared to a reference measured with the same instrument. While this is true in first order approximation, the authors correctly note that most DOAS retrievals do allow for a small shift and/or squeeze of the spectrum when compared to the reference to account for temperature-induced changes of the spectrometer calibration, thus foiling the 'canceling out' of the variable sensitivity.

The authors go on to show that it is actually possible to model the collected moderate resolution UV spectra using a high-resolution solar reference spectrum collected with a different instrument of much higher resolution. The concept here is quite similar to the standard practice of convolving and down-sampling reference absorption cross-sections of trace gases measured with high-resolution spectrometers in the laboratory. From a practical standpoint, this is a very useful application as it eliminates the need for recording 'clear sky' reference spectra, at least when an intensity precision of a couple of percent is sufficient, for example when measuring volcanic plumes with more than 20 ppm.m of SO$_2$ in the measured column.

On the other hand, a few of the concepts discussed in the manuscript appear flawed or at least unclear. For one, the authors put a lot of emphasis on the fact that the iFit procedure performs the spectral fit in the intensity space as opposed to the optical depth space. To me, this seems to be a more or less arbitrary choice and it remains unclear why one is advantageous over the other. Performing the fit as is conventionally done in optical depth space requires converting the measured intensity spectra to optical depth by taking the logarithm. But performing the fit in intensity space first requires converting the

column densities of the retrieved trace gases to intensity space by taking the exponent. I do not see a computational advantage here, except in the case of strong absorption (see specific comments below) and though the ability to take non-linear radiative transfer effects into account is hinted at, there's no way to assess the merits of this claim without a better description of the actual process.

It appears to me that the fit itself is not as different from the conventional DOAS method as the manuscript makes it sound. The paper lacks a mathematical formulation of the fit procedure. Given that the manuscript is focused on the iFit methodology, this is clearly needed. Once explicitly formulated, it will become clearer what the exact differences are between iFit and the conventional DOAS method. In fact, it would be good to do a side-by-side comparison of the two, both in theory and in application. Also, a significant number of additional references are required throughout, most of which are given by Reviewer 1.

The discussion of errors remained unclear to me. The measurement errors refer to the uncertainty of the retrieved trace gas column densities. If the fit is performed in intensity space, then the fit errors would refer to errors in measured intensity, not $SO_2$ column density. In this point the iFit approach seems like it might be at a disadvantage when compared to the conventional DOAS fit, where errors in optical depth are linearly related to errors in the retrieved column density. Also, the actual measurement error seems not to be defined by the noise, but rather dominated by the inaccuracy of the procedure, as evidenced by the dependency of the results on the lower limit of the fit window.

In conclusion, there are a number of interesting concepts introduced in this manuscript. However, the main focus of the manuscript is the fitting in intensity space (iFit) which is not described well enough to ascertain where its strengths truly lie. Is it really so much different than conventional DOAS? In which cases is it different? When is one method better than the other? All of these questions remain unanswered.

**Specific Comments**

L83 – The authors state that the broadband component of scattering and absorption is removed in the DOAS approach. This is not necessarily true. Many modern DOAS retrievals do not high-pass filter the absorption cross-sections, and instead include a polynomial (typically order 3 – 5, see Eq 4 below) in the optical depth fit to account for broadband changes in light transmittance. It is true that the information captured by this polynomial is usually not used for future analysis, and in that sense it is simply used to 'remove' the broadband component. However, the polynomial parameters essentially contain the exact same information as those of the iFit polynomial. I don't see a fundamental difference in the retrievals here, unless the authors can show how the iFit polynomial can be used in a way that the DOAS polynomial cannot be.

L99 ff – The DOAS approach described here is one very specific implementation of a DOAS fit that does indeed remove the broadband component from the spectra before applying the fit. However, this is not a common methodology. It's probably best to refer to Platt and Stutz (2008) for the common DOAS fit. Their section 8.3 is titled "DOAS Analysis Procedure". Here they describe the following method:

Starting from the Beer-Lambert-Law

$$I = I_0 \cdot e^{-\tau} \qquad\qquad \text{Eq 1}$$

where I is the measured intensity ('plume spectrum'), $I_0$ is the initial intensity ('clear sky' or Fraunhofer spectrum), and $\tau$ is the optical depth:

$$\tau = -\sum_i \sigma_i \cdot S_i + \varepsilon_R + \varepsilon_M \qquad\qquad \text{Eq 2}$$

Here, $\sigma$ are the absorption cross sections, S are the column densities, and $\varepsilon$ are the extinction coefficients for Rayleigh and Mie extinction.

The Beer's law can be conveniently rewritten as

$$lnI = lnI_0 - \tau \qquad\qquad \text{Eq 3}$$

A common forward model used in DOAS actually uses this equation and attempts to reproduce the logarithm of the measured 'plume spectrum' with a linear model:

$$lnI \approx lnI_0 - \sum_i \sigma_i \cdot a_i + R + P_\lambda \qquad\qquad \text{Eq 4}$$

Here, R is a Ring correction spectrum, $a_i$ are forward model parameters that are varied to find a best match between model and measurement (these later give the respective column densities), and $P_\lambda$ is a polynomial used to describe broadband structures. In this simple form, the polynomial will capture the same broadband structures that the proposed iFit polynomial would capture.

L119 ff – There are two main reasons why the simple DOAS method becomes inaccurate for strong absorption, but this is not one of them. DOAS does not assume a linear relationship between the optical depth $\tau$ and $I_m/I_{sky}$. Instead, DOAS assumes a linear relationship between $\tau$ and $ln(I_m)$, or between $\tau$ and $ln(I_m/I_{sky})$, as shown in Eq 3 above. This relationship holds true even for very high optical depths. No, the problem lies in two second-order effects:

1) The measurement is performed at insufficient resolution to fully resolve the absorption lines. In effect, the instrument convolves the absorption bands with the instrument line shape, and this occurs in intensity space. In cases of strong absorption, the absorption cross-sections used for the retrieval cannot simply be convolved in optical depth space (as shown in your Equation 5). More information can be found in Platt and Stutz (2008), section 6.7.2 Application of the DOAS Approach in Practical Situations, Case 3: Strong Absorbers at Low Resolution and a Smooth Light Source. Dealing with this issue does appear to be a little easier in intensity space, though the

model is still non-linear as the parameters we are interested in (the column densities) are still exponentially related to the measured intensity. The iFit might have a slight advantage in computational complexity here, since the computationally expensive convolution operation only needs to be performed once in each fit iteration, rather than separately for each individual absorption cross-section. But it is unclear how much this really improves processing speed. And this is only relevant for strong absorption.

2) In scattered light spectroscopy, the effective path that the radiation has taken on its way to the instrument becomes dependent on wavelength when strong absorption bands are encountered. At wavelengths of strong absorption, the effective path length is typically reduced as the probability for long paths is decreased. Neither the conventional DOAS nor the iFit approach can easily take this into account – at least it hasn't been shown in this manuscript.

L151 – 'Correct application of the DOAS approach requires all calculations of fitted spectra are first performed in transmittance…'. This is only true for strong absorbers. For weak absorbers, Eq 4 (above) provides a very simple model that is linear and thus much easier to solve.

L160 – iFit cannot take into account the wavelength dependency of the photon path length any more than conventional DOAS. At least you haven't shown how.

L163 ff– These three motivations are unclear to me. I agree that fitting the literature solar reference spectrum is advantageous, but this could easily be implemented in optical depth space too, or not? Fewer manipulations are better, but taking the logarithm is pretty straight-forward, and the iFit instead has to take the exponential of the absorption cross-sections, so which is better? The final point (regarding low spectral intensities at low solar zenith angles) is not clear to me. I think it can be avoided by implementing the conventional DOAS fit as described in Equation 4 above, right?

L 194 – What aspect of iFit is specific to Ocean Optics USB2000 spectrometers?

L204 – Can you give a reference for the combination of Gaussian and boxcar to describe the instrument line shape? How does the manual adjustment work?

L307-310 – It appears to me that the spectral resolution of the literature reference cross-sections could be important in the iFit approach. Are all these cross-sections measured at 0.01nm resolution or better?

L413 – 'The validity of the approach is demonstrated…' It is not clear to me that this is a valid statement. Since an average residual is being included in the fit, wouldn't one expect the result to be pure noise? What else could it be? Any processes not properly accounted for by the forward model are being removed by fitting the residual. Incidentally, the authors attribute the systematic structures in the residual to an imperfect solar reference spectrum, but there may also be other explanations. What about the presence of unknown trace gases in the boundary layer? Their absorption should also be approximately independent of solar zenith angle.

L503 – '…successfully included all features…'. Yes, this will always be the case if the residual is included in the fit, right?

L509 and figure 5 – It appears that the y-error (plotted on the vertical axis in figure 5) was calculated by dividing by a boxcar smoothing rather than a polynomial fit as was previously mentioned in the manuscript. Either way, I don't understand why the y-error would change. Shouldn't it be the same, regardless of which fit method was used? But the figure indicates lower y-errors for the 'flat removed' and 'residual removed' fit methods when compared to the basic iFit. Please explain.

L552 and figure 9 – What do you mean by 'the empirical precision'? Is that the precision derived from the measurement noise? Somewhat alarmingly, there appears to be a 25% difference in retrieved $SO_2$ columns when moving the lower fit limit from 305 to 314 nm. Where does this come from? I wonder if it might be caused by inadequate correction of stray light in the spectrometer. I can think of no other explanation why this would occur. What does this look like for a conventional DOAS fit?

L595 – 'the quality of the fit is limited by measurement noise…'. What does this actually mean? The fit precision? The measurement accuracy seems to depend quite severely on the lower wavelength of the fit window. This statement seems to imply that the quality of conventional DOAS fits is in some way compromised, but this has not been shown in the manuscript, except perhaps in the case of strong absorbers (but there are known solutions to this problem too, see reference above to Platt and Stutz 2008).

L599 – 'Intensity-based retrievals also allow direct characterization of aerosol/ash absorptions and accurate modeling of light dilution effects…' This was not shown in the manuscript, so the statement is impossible to asses. However, I cannot see any clear advantage of the iFit in these respects, other than perhaps a slightly easier implementation, but this is probably a marginal benefit at best, and was also not shown in the manuscript.

L603 – I would argue that the 'flat correction' and use of a literature solar spectrum open up new possibilities, not the fitting in intensity space. Flat correction and ensuing use of the Kurucz spectrum are very good ideas indeed!

L655 – The polynomial actually takes into account more than just aerosol absorption – it accounts for Rayleigh scattering, aerosol scattering, aerosol absorption, broadband absorption of unknown trace gases and broadband changes in the spectral sensitivity of the instrument.

L657 – Given the previous comment, it is really unclear how this polynomial will be used to characterize interference in $SO_2$ camera measurements. Please explain or omit this point. Also give a reference for the $SO_2$ camera technique. Also apply your changes to L687.

**Minor issues**

L 16 – The iFit fitting procedure could be useful in a wide range of applications. Perhaps it's better to say that volcano monitoring is just one of many potential applications.

L28 – Again, this could probably be applied to trace gases other than $SO_2$ and BrO too.

L43 –The USB2000 spectrometer is a specific model made by Ocean Optics. While it may have been the first such miniature spectrometer, there are now many others on the market, so it is probably best to avoid using a single make and model throughout the manuscript.

L81 – Include a reference for 'light dilution', e.g. Kern et al 2009

L 197 – I think the FLYSPEC doesn't use a fiber. This is not a requirement in any case.

L216 – The dark current also may contain 'hot pixels' that may not be insignificant.

L221 – What happens if the bias level falls below zero?

L235 – This assumes that the stray light is constant across the detector. Is it? I think some people use a first order polynomial, but I'm not sure which approach is better.

L301 – Please rephrase this sentence. I know what you mean, but it's misleading because the Ring effect is not related to the absorption processes responsible for the structured solar spectrum.

L304 – Please give a reference for your method of deriving the Ring correction spectrum.

L324 – 'Gas amount'. I think you mean column density?

L327 – This description of the fit method is still a bit unclear. How is the polynomial actually calculated? I assume the polynomial parameters are included as variable parameters in the fit? Please add an explicit, mathematical description of how the forward model as well as the procedure for each fit iteration.

L446 – 'The CCD sensor…' This paragraph repeats information given earlier on in the manuscript. Consider removing.

Figure 1 – The figure caption indicates that boxcar smoothing was used to normalize the spectra, not polynomial fitting as mentioned in the text.

**Concluding remarks**

Three important things are really missing from this manuscript:

1) an explicit mathematical description of the forward model and the iFit iterative method
2) a comparison to conventional DOAS methods such as the one given in Eq 1-4 above
3) a discussion of the advantages and disadvantages of the two methods, and a description of the situations in which one may be better than the other.

In its current state, the manuscript has not convinced me that the iFit approach is superior to conventional DOAS. It appears that there might be some advantages in cases where strong absorption is encountered, where the convolution of absorption cross-sections cannot be performed in optical depth space. In cases of weak absorption, however, the conventional DOAS fit seems advantageous because

the model for describing ln(I) is linear, so the fit is linear (except when adding a spectral shift and squeeze, but Beirle et al (2013) describe a way to linearize even this problem).

Also, it should be noted that neither iFit nor conventional DOAS take into account the effect of variable light path length due to radiative transfer issues. For this, one would still need a fit coupled to a radiative transfer model such as described by Yang et al (2010) or Kern et al (2012). Several sections of the manuscript are misleading in this point.

The concept I found most interesting in this manuscript is the correction of the variable sensitivity across the detector with a 'flat spectrum', and the resulting ability to fit a reference solar spectrum from literature rather than a measured Fraunhofer spectrum. This concept would be widely applicable to absorption spectroscopy of the atmosphere. However, couldn't one argue that this technique can just as easily be applied to a conventional DOAS fit as it can to iFit if one simply puts the logarithm of the literature Fraunhofer reference into Eq 5 as $\ln(I_0)$? In any case, these concepts are novel and actually quite useful, so I recommend that the manuscript be reconsidered for publication in Atmospheric Measurement Techniques once the aforementioned concerns are addressed.

**References**

Beirle S, Sihler H, Wagner T (2013) Linearisation of the effects of spectral shift and stretch in DOAS analysis. Atmos Meas Tech 6:661–675. doi: 10.5194/amt-6-661-2013

Kern C, Deutschmann T, Vogel L, Wöhrbach M, Wagner T, Platt U (2009) Radiative transfer corrections for accurate spectroscopic measurements of volcanic gas emissions. Bull Volcanol 72:233–247. doi: 10.1007/s00445-009-0313-7

Kern C, Deutschmann T, Werner C, Sutton AJ, Elias T, Kelly PJ (2012) Improving the accuracy of SO2 column densities and emission rates obtained from upward-looking UV-spectroscopic measurements of volcanic plumes by taking realistic radiative transfer into account. J Geophys Res 117:D20302. doi: 10.1029/2012JD017936

Yang K, Liu X, Bhartia PK, Krotkov N a., Carn S A., Hughes EJ, Krueger AJ, Spurr RJD, Trahan SG (2010) Direct retrieval of sulfur dioxide amount and altitude from spaceborne hyperspectral UV measurements: Theory and application. J Geophys Res 115:1–15. doi: 10.1029/2010JD013982

---

## Author Comment (AC1) · 14 Jul 2016

The reviewer's overall positive assessment, and meticulousness, is appreciated and has resulted in an improved revised manuscript. The key comments on required improvements in the mathematical presentation is clearly understood and will be addressed through revisions to the equations and improved figures to allow reproduction by the community. The intercomparison with other techniques is desirable but is outside the main focus of the paper which is, as the reviewer highlights, to present an approach which is free from a measured reference spectrum. It would be possible to give egregious examples of where the true SO2 value was miscalculated due to the use of a contaminated reference spectrum, but this is self-evident. As this is the first

order objective of the paper, this remains the focus, and no direct intercomparison is presented in the revised document. To reflect this, the focus of the paper is revised to exclusively focus on a solution to the reference spectrum problem, with no further claims to improvement over the DOAS approach arising from the intensity spectrum approach. A thorough intercomparison between intensity based fitting and DOAS fitting will be done in a further work.

In response to General comments:

1. As discussed above, the focus of the revised manuscript is on the reference spectrum calculation, and the positive comments on the advantages of the intensity fitting over DOAS have been removed, and addressed in a further intercomparison work.

2. Runtimes are now discussed, and on a modern PC very acceptable.

3. A revised mathematical description has been included.

4. Light dilution comments have been deleted.

5. Language has been tightened up throughout.

6. While the reviewer's use of language such as 'very roughly approximated' is perhaps also suffering from the weakness highlighted in comment 5, we highlight that these residuals are significantly smaller than those produced by neglecting the impact of the flat spectrum. Further details of this residual behaviour and its stability in time and space has been included.

In response to specific Comments:

Points 1-20 Suggestions accepted

Point 21: Ring spectrum could be improved in a final operational version of the algorithm, but as the purpose of this paper is to highlight an effective solution to the reference spectrum calculation the relatively rough but adequate RRS solution presented here is sufficient to address the purpose of the paper. Further developments

will improve on this point in future work.

Point 22. The pixel by pixel QE of the spectrometer is exactly what is being addressed by flat spectrum characterisation, so this is not the source of the issue.

Point 23. The manuscript has been updated to show that the residual is stable in time and space and therefore offers an effective general solution.

Point 24. Language adjusted.

Point 25. ILS fitting has been described better to reflect the approach used.

Point 26: Optimal estimation was used with no apriori information, and therefore it is effectively a standard non-linear solver.

Point 27, 28 corrected

Point 29. A new K is calculated at each iteration.

Point 30, 31 Clarified

Point 32. I show a lack of temperature dependence for the flat spectrum. Dynamic fitting of the ILS can be used to address a T-dependent ILS. This is beyond the scope of the current work.

Point 33. Improved language

Point 34. Not addressed, there is now plentiful referencing of the relevant literature.

Point 35. If the reviewer had a reference I would use it.

Point 36, 37, 38. The language has been modified.

Point 39 Done

Point 40. Nothing to address

Point 41. Interesting comments, but again the main focus here is on the artificial refer-

ence spectrum stability.

Point 42. Done

Point 43. If the BrO was not volcanic there would be not be a correlation with SO2, notwithstanding the reviewer's assertion to the contrary, as the volcanic signal is changing rapidly and randomly,

Response to technical comments:

1,2 done
* * *

---

## Author Comment (AC2) · 14 Jul 2016

Author Comment to review AMT-2015-380-RC2

We appreciate the reviewer's overall positive assessment of our manuscript. Through addressing the points the reviewer has raised we feel the manuscript has been significantly improved. The main difference in the revised manuscript is that we emphasise the contribution of this paper as offering a solution to the reference spectrum issue, with the advantages of the intensity fit somewhat de-emphasised.

Line 83: Agreed, this point has been deprecated

Line 99: Thanks, changed

[Figure]

Line 119: Thanks, changed

Line 151: Changed, but highlighted that volcanic SO2 signals are often »0.1 OD

Line 160: Corrected

Line 163, 194: Adjusted

Line 204: Adjusted and improved

Line 307-310: Clarified

Line 413, 503: We test the stability of the approach more thoroughly in the revised manuscript

L.509: clarified

L.552, The lower limit sensitivity has been examined for the source of the 25% error and this has been clarified.

L595: Language adjusted

L.599: Deleted

L603: Yes, we have adjusted the whole manuscript to highlight these ideas and diminish the intensity fitting somewhat

L665: Corrected

L657: Deleted

Minor issues: All addressed.